# “Paralympic Brain”. Compensation and Reorganization of a Damaged Human Brain with Intensive Physical Training

**DOI:** 10.3390/sports8040046

**Published:** 2020-04-07

**Authors:** Kimitaka Nakazawa, Hiroki Obata, Daichi Nozaki, Shintaro Uehara, Pablo Celnik

**Affiliations:** 1Department of Life Sciences, Graduate School of Arts and Sciences, The University of Tokyo, Tokyo 1538902, Japan; 2Department of Humanities and Social Sciences, Institute of Liberal Arts, Kitakyushu Institute of Technology, Kitakyushu 8048550, Japan; obata@dhs.kyutech.ac.jp; 3Department of Education, Graduate School of Education, The University of Tokyo, Tokyo 1130033, Japan; nozaki@p.u-tokyo.ac.jp; 4Department of Physical Medicine and Rehabilitation, Human Brain Physiology and Stimulation Laboratory, Johns Hopkins University, Baltimore, MD 21287, USA; suehara@fujita-hu.ac.jp (S.U.); pcelnik@jhmi.edu (P.C.); 5Japan Society for the Promotion of Science, Tokyo 1020083, Japan

**Keywords:** Paralympic sports, case study, female, motivation, electromyography (EMG), swimming, neurorehabilitation

## Abstract

The main aim of the study was to evaluate how the brain of a Paralympic athlete with severe disability due to cerebral palsy has reorganized after continuous training geared to enhance performance. Both corticospinal excitability of upper-limb muscles and electromyographic activity during swimming were investigated for a Paralympic gold medalist in swimming competitions. Transcranial magnetic stimulation (TMS) to the affected and intact hand motor cortical area revealed that the affected side finger muscle cortical representation area shifted towards the temporal side, and cortico-spinal excitability of the target muscle was prominently facilitated, i.e., the maximum motor evoked potential in the affected side, 6.11 ± 0.19 mV was greater than that in the intact side, 4.52 ± 0.39 mV (mean ± standard error). Electromyographic activities during swimming demonstrated well-coordinated patterns as compared with rather spastic activities observed in the affected side during walking on land. These results suggest that the ability of the brain to reorganize through intensive training in Paralympic athletes can teach interesting lessons to the field neurorehabilitation.

## 1. Introduction

It is well established that the human central nervous system has the capability to reorganize after focal lesions like stroke and spinal cord injury, if physical rehabilitation is applied properly according to the types of disabilities [1,2]. While athletic training that Paralympic athletes engage in is not regarded as physical rehabilitation since the goal is to improve their performance and increase their competitive level rather than to improve physical functions for daily living, this athletic training is also known to induce plastic changes in the central nervous system in a use-dependent manner [3,4,5]. This athletic training, therefore, is also expected to induce central nervous system (CNS) reorganization in the Paralympic athletes.

However, there seems to be another major difference in athletic training and rehabilitation, that is, the level of motivation that athletes and patients have. It is quite certain that every athlete has high levels of motivation to win a game or to improve their performances, while in patients after physical damage it is difficult to keep motivation up or even to participate in rehabilitation programs [6]. Although motivation is well recognized to affect neural plasticity and it in turn has a great impact on rehabilitation outcomes [7,8], how and to what extent motivation modulates neural plasticity is still obscure.

The para-athletes, therefore, can be excellent examples that show how, and to what extent, the human brain reorganizes, if intensive physical training is provided continuously to the human body with disabilities with higher motivation levels [9,10]. In the present study, we explored the brain and neuromuscular activities of a Para-gold medal swimmer who has a hemiplegic type of paralysis in her left side body. The main aim of the presented study was evaluation of brain–muscle connectivity as well as her neuromuscular activities both during swimming in water and walking, to provide new insight into neurorehabilitation studies, namely, “Paralympic brain”.

## 2. Materials and Methods

### 2.1. Participant

The participant of this study was a Paralympic swimmer ZZ who has won totally 12 medals including Gold awards in the Paralympic games of Beijing (2008), London (2012), and Rio de Janeiro (2016). ZZ, a 25-year-old female, suffers from hemiplegic cerebral palsy (CP). She has moderate levels of paralyses in her left side limbs. She is ambulant with a dropped foot in the affected side, and her left elbow joint is flexed at about 90 degrees due to hypertonia in the elbow flexors. Independent finger movements in the affected side are impossible due to moderate to strong muscle tone. She started swimming when she was seven years old. After she got involved in Paralympic swimming, she has continued regular training, that is, swimming for two to four hours a day, with resistance training sometime in between the swimming training. Figure 1 shows MRI images of her brain and the lesion site in black.

The participant received a detailed explanation of the experimental procedures before the experiment and provided written informed consent. The study was approved by the Ethical Committee of the Graduate School of Arts and Sciences (No 475-4), The University of Tokyo. The experiment was carried out according to the principles and guidelines of the Declaration of Helsinki (1975).

### 2.2. Experiments

#### 2.2.1. Experiment 1

In this experiment, we evaluated corticospinal excitability of distal upper-limb muscles as a measure of brain (i.e., primary motor cortex; M1) reorganization using transcranial magnetic stimulation (TMS) techniques.

1.Transcranial magnetic stimulation

We applied single-pulse stimulation over the right (lesioned hemisphere) and left (nonlesioned hemisphere) primary motor cortex (M1) using a 70 mm-diameter figure-of-eight coil connected to a Magstim 200^2^ stimulator (Magstim Co. Ltd, Whitland, UK). We first identified the “hotspot” in both hemispheres as the optimal area for eliciting motor-evoked potential (MEP) in the contralateral resting first dorsal interosseous (FDI) muscle. The coil was placed tangentially to the scalp with the handle pointed backward at a 45° angle with respect to the anteroposterior axis. To clarify the positional relationship between the brain and the coil, the participant’s T1-weighted anatomical images (170 slices; voxel size: 1.0 mm × 1.0 mm × 1.2 mm; repetition time (TR): 8 ms; field-of-view (FOV): 256 mm × 256 mm; flip angle (FA): 8°; echo time (TE): 3.8 ms) scanned with a 3.0-T Achieva 3.0 scanner (Phillips, Best, The Netherland) were imported into a frameless neuronagvigation system (BrainSight, Rogue Research Inc, Montreal, QC, Canada), in which we marked stimulation sites over the anatomical image. Then, in this optimal spot, we determined the resting motor threshold (rMT) as the minimum TMS intensity (maximum stimulator output; MSO) that evoked MEPs of 50 μV in 5 of 10 trials at rest. Muscle relaxation was monitored by providing visual feedback of electromyographic (EMG) signal. The rMT was 40% and 44% of MSO in the right and left hemispheres, respectively.

2.MEP recruitment curve (stimulus response curve)

We assessed MEP responses relative to TMS intensity changes to evaluate corticospinal excitability of the FDI both in the lesioned and nonlesioned hemispheres. We delivered TMS pulse over the FDI hotspot in each hemisphere. The TMS intensity range was set from 110% to 150% of rMT in steps of 10%. We recorded 10 MEPs at each intensity step with an interstimulus interval of 5 seconds and calculated the average peak-to-peak amplitude for each intensity. Electromyographic activity was recorded from the FDI muscles in both sides using surface electrode pairs and pregelled disposable Ag/AgCl electrodes (A10040, Vermed Inc., Buffalo, NY, USA) placed on the skin, about 3 cm apart in a belly-tendon montage. Signals were amplified (×1000; AMT-8 EMG, Bortec Biomedical Ltd., Calgary, AB, Canada), sampled (1 kHz) with a band-pass filter of 10–1000 Hz, and recorded with a data acquisition unit CED 1401 controlled by Signal 5 Software (Cambridge Electronic Design, Cambridge, UK). We analyzed EMG signals off-line using MATLAB (R2015b; MathWorks, Natick, MA, USA).

#### 2.2.2. Experiment 2: Recording of Electromyographic (EMG) Activities during Swimming

It was visibly apparent that the ZZ’s affected side upper limb can move more dynamically in water as compared to on land. In order to examine if there are any differences in EMG activities during swimming in water and motor activities on land, we chose walking as a locomotive motor activity on land for the comparison. In Experiment 2, two different motor tasks were performed by subject ZZ while we recorded EMG activity: 1) walking at a comfortable speed and at a faster speed, and 2) swimming in breast stroke and free style at comfortable and faster speeds. The reason why faster speeds were tested in addition to the comfortable speed was that we aimed to make expected differences in EMG activities during swimming and walking much clearer. For the walking test, she walked straight for about 40 m on the pool-side at the two different speeds, and for the swimming test she swam for 50 m at the two different speeds. Each measurement trial was repeated two times. 

To quantify EMG activity levels while performing each motor task, the maximum voluntary activation levels were tested manually for each muscle. EMG activities from a total of 16 muscles were recorded using a wireless system (Free EMG, Milan, Italy) while she walked on the floor and swam in a standard pool. The muscles tested were the biceps brachii (BB), triceps brachii (TB), flexor carpi radialis (FCR), extensor carpi radialis (ECR), rectus femoris (RF), biceps femoris (BF), soleus (SOL), and tibialis anterior (TA), from both side limbs. Root mean square (RMS) values of EMG signals for five seconds during stable phases in the walking and swimming tasks were obtained to compare with those values during the maximum voluntary contractions of each muscle. 

3.Statistics

No statistical comparisons were performed in this study, since this is the single case study. Data were presented basically as means and standard errors (s.e.).

## 3. Results

### 3.1. TMS and MEP Recruitment Curves

The yellow dots in Figure 1 indicate the hot spot positions where the TMS induced the clearest MEPs in each side FDI muscle. The hot spot positions of both sides first dorsal interosseous were not symmetrical in the left and right hemispheres, indicating upward shift of the right hemisphere hot spot.

Figure 2 shows the changes in MEP amplitude in relation to the applied TMS intensity (% of resting motor threshold, rMT) for both side FDI muscles. The results showed that the maximum slope of the sigmoidal curve (0.99 vs. 0.28 mV/%rMT) and the maximum MEP amplitude (6.11 ± 0.19 mV vs. 4.52 ± 0.39 mV, mean ± s.e.) were higher, and the motor threshold was lower (44 vs. 48 % of maximum output of the device) in the affected side FDI than in the nonaffected side FDI. This indicates that the corticospinal excitability of the affected side FDI was higher than that of the nonaffected side FDI.

### 3.2. EMG Activities during Walking and Swimming

Figure 3 and Figure 4 show typical examples of EMG activities recorded during walking at a comfortable speed and swimming in a pool. As observed in this figure, EMG amplitude in the affected limbs increased overall and showed clear on–off phases during swimming, unlike that during walking on land. This tendency was more clearly observed in the affected side of upper limb muscles. As noticed from her hemiplegic type of walking posture when she walked on land, aberrant EMG bursts appeared irregularly in the affected upper limb muscles. In water, however, the greater and rhythmic EMG bursts appeared, which corresponded to the dynamic upper limb motions during swimming. 

## 4. Discussion

The results of the present study revealed that 1) motor cortical area, specifically the area innervating the left FDI muscle was drastically reorganized, and 2) the EMG activities in the affected side during swimming were greater in size and qualitatively well-coordinated compared with those recorded during functional movements on land. These results suggest that the swimming training that has been performed by ZZ since the age seven years old has likely played a major role in the observed reorganization of her brain, resulting in well-coordinated muscular activities in water. 

The supposed underlying neural mechanisms are discussed below.

### 4.1. Reorganization of Motor Cortical Area

As shown in Figure 1, FDI motor cortical areas of both sides identified with MEPs by TMS demonstrated that the hot spot in the injured hemisphere is located in a more medial region than that of the contralateral nonaffected hemisphere. This means that the FDI cortical representation has shifted to the area innervating trunk or lower limb muscles in the affected side of the brain. More surprisingly the corticospinal tract (CST) excitability assessed by the input/output relation in Figure 3 clearly demonstrated that the CST excitability of the affected side was significantly higher than that of the nonaffected side. This finding differs from that in adult stroke patients who often demonstrate a flatter input/output recruitment curve and less CST excitability in the affected side than the less-affected side [11]. Although underlying mechanisms leading this difference remain unclear, in persons with cerebral palsy, depending on the developmental stage, size, and location of the brain damage, the CST development during maturation process is known to undergo various pathophysiological changes [12]. 

In the case of ZZ, she has been involved in normal rehabilitation for cerebral palsy as well as swimming training since she was seven years old. Both rehabilitation and swimming training might affect the motor cortical reorganization observed in the present study. Specific motor skill training is known to change motor cortical representation and its excitability [13,14,15]. These changes are more prominent in the brains of professional athletes and musicians who have been engaged in higher level motor skill training for a long time [4,5,16,17]. In the case of a Paralympic long-jumper with below-knee-amputee, Mizuguchi et al. (2019) [9] found the unique bilateral motor cortical activation when he exerted static knee muscle force in the amputated side of lower limbs, which was not observed in the other subjects with below knee amputee who were not involved in any sports activities. These previous studies suggest that the long-term swim training has induced long-term use-dependent plastic changes that resulted in reorganization of her brain in addition to the expected changes through development and physical rehabilitation.

### 4.2. The Well-Coordinated Muscular Activities in Water

The results of the present study showed that the recorded EMG activities in the affected side of ZZ were greater in size and showed clearer on–off phases during swimming than walking on land. These characteristics were shown both quantitatively and qualitatively, indicating her well-coordinated movements in water rather than on land. This was visually apparent when one saw her swimming, which was performed with more dynamic upper limb movements. Furthermore, as shown in Figure 5, in many muscles in the affected side the relative EMG sizes to the MVC levels were greater than 100%, which means that the EMG activation levels of those muscles during swimming were increased to a higher level than the maximum voluntary activation levels on land. The most conceivable interpretation is that the better motor coordination of ZZ in water is likely due to her intense swim training that she has been doing since early childhood.

The neural mechanisms underlying the higher motor functions of ZZ in water is likely due to cortical reorganization akin to use-dependent plasticity and reinforcement mechanisms induced by the long-term repetitive nature of swim practice.

Of note, other factors should be taken into account, such as activity of the autonomic nervous system. It is well known that physical characteristics of water have a reduction effect on sympathetic nervous system activity [18]. For instance, there is experimental evidence indicating that increased sympathetic activity enhances gamma drive, which in turn increases spinal stretch reflex excitability [19,20]. The hyper stretch reflex excitability is known to be the main neural factor underlying spastic hypertonia, which is typically observed in persons with cerebral palsy. The observed flexed position of elbow joint in the affected side of Subj. ZZ on land indicates the presence of spastic hypertonia in the elbow flexor muscles. Thus, the lower sympathetic drive present in water may reduce her spastic hypertonia in the affected side, facilitating better muscle coordination than when walking over ground. Furthermore, her having less fear of falling in water is supposedly due to her well-coordinated swimming movement. A higher fear of falling is known to augment sympathetic nervous activity, and as a result it increases stretch reflex excitability via augmented muscle spindle sensitivity [21]. Supposedly, ZZ has no fear of falling in water as she expressed “I am free in water”, meaning free from gravity. It can be assumed therefore that the combination of a sort of relaxation effect of water and her diminshed fear of falling further decreased her sympathetic activity in water. This would allow her physical movements in water to be more flexible and well-coordinated while eliminating hypertonia. Under such a circumstance in water, she could have learned and obtained well-coordinated swimming skills through extensive daily training from her childhood.

Finally, the results in the present study may have great impact in neurorehabilitation, especially for cerebral palsy, since it shows that she can move her affected side limbs in water with well-coordinated muscular activities, which possibly could not be accomplished with conventional rehabilitation. This is in line with the specificity effects of motor learning with diminished generalization to other tasks. Future studies are expected to explore underlying neural mechanisms of these processes in more detail.

In conclusion, this case report demonstrates the evidence of significant cortical and muscular reorganization following cerebral palsy in a Paralympic gold medalist swimmer. The findings exposed the magnitude of brain plasticity following a brain lesion, which leads to a remarkable improvement in motor abilities when the person is highly motivated and trains very intensively.

## Figures and Tables

**Figure 1 sports-08-00046-f001:**
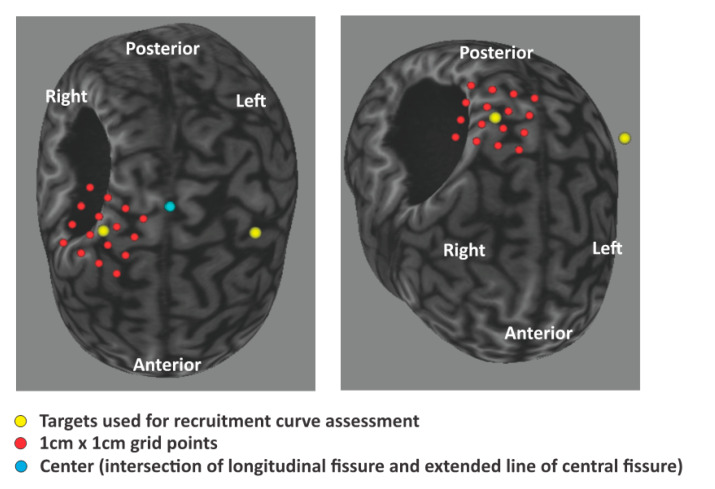
Brain images and transcranial magnetic stimulation (TMS) targets indicated in red and yellow dots. As shown in this brain image, the black colored area is the large damaged area, most probably due to a stroke suffered perinatally. Note the yellow dot in the right hemisphere locates more temporally than the yellow dot in the left hemisphere.

**Figure 2 sports-08-00046-f002:**
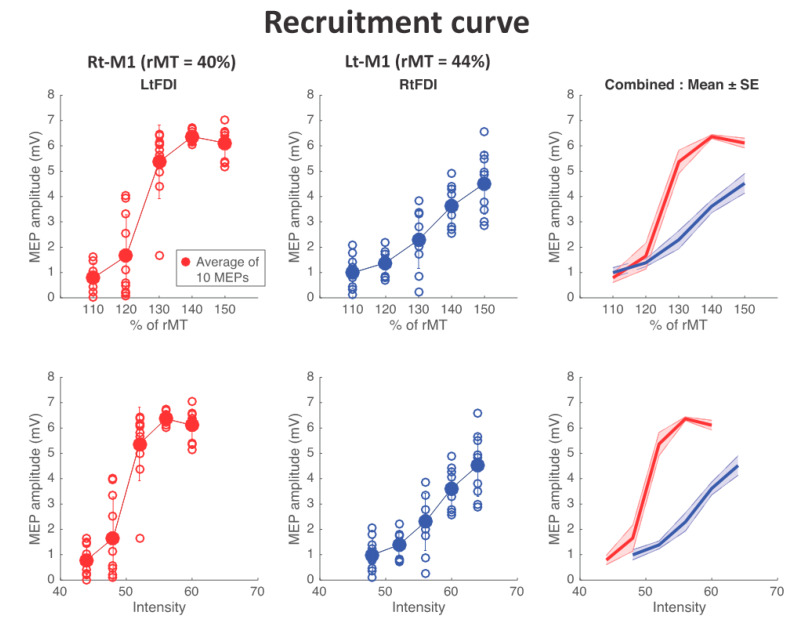
Relations (recruitment curve) between the TMS intensity (%) and motor-evoked potential (MEP) sizes obtained from first dorsal interosseous in the affected (red) and nonaffected (blue) sides. Each open circle indicates the amplitude of MEP to the single TMS, and the filled circles indicate the average of ten MEPs at each TMS intensity. The right panels show comparisons between the affected side (red) and nonaffected side (blue) recruitment curves. Note the abscissa of the upper panels is normalized with the motor threshold (MT) intensity, while the abscissa of the lower panels is the absolute output intensity (% of the maximum output of the TMS apparatus). The higher slope, the maximum MEP levels (plateau level in the LtFDI), and the lower MT clearly demonstrate that excitability in the corticospinal tract to the affected side first dorsal interosseous (FDI) is much greater than that of the nonaffected side.

**Figure 3 sports-08-00046-f003:**
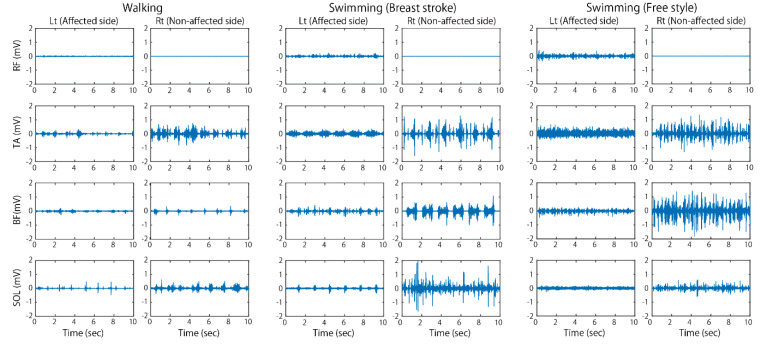
Electromyographic (EMG) activities in lower limb muscles recorded during walking on land and swimming in water. Note the increased EMG amplitude in the affected side muscles during swimming as compared to EMGs during walking.

**Figure 4 sports-08-00046-f004:**
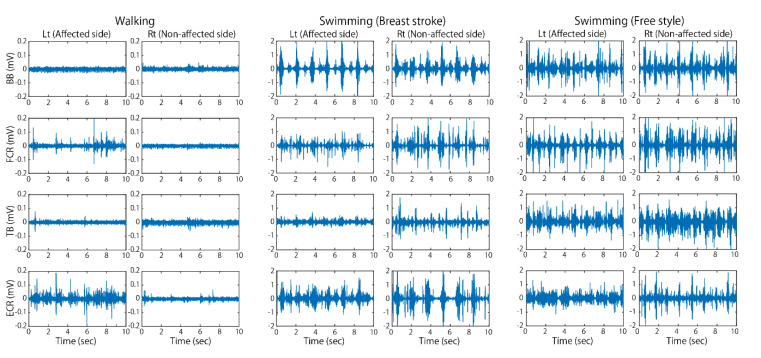
EMG activities in upper limb muscles recorded during fast walking on land and fast swimming in water. The clear phasic and rhythmic EMG activities appeared in the affected side of muscles during swimming, while during walking rather aberrant bursts were observed in the wrist muscles, FCR, and ECR.

**Figure 5 sports-08-00046-f005:**
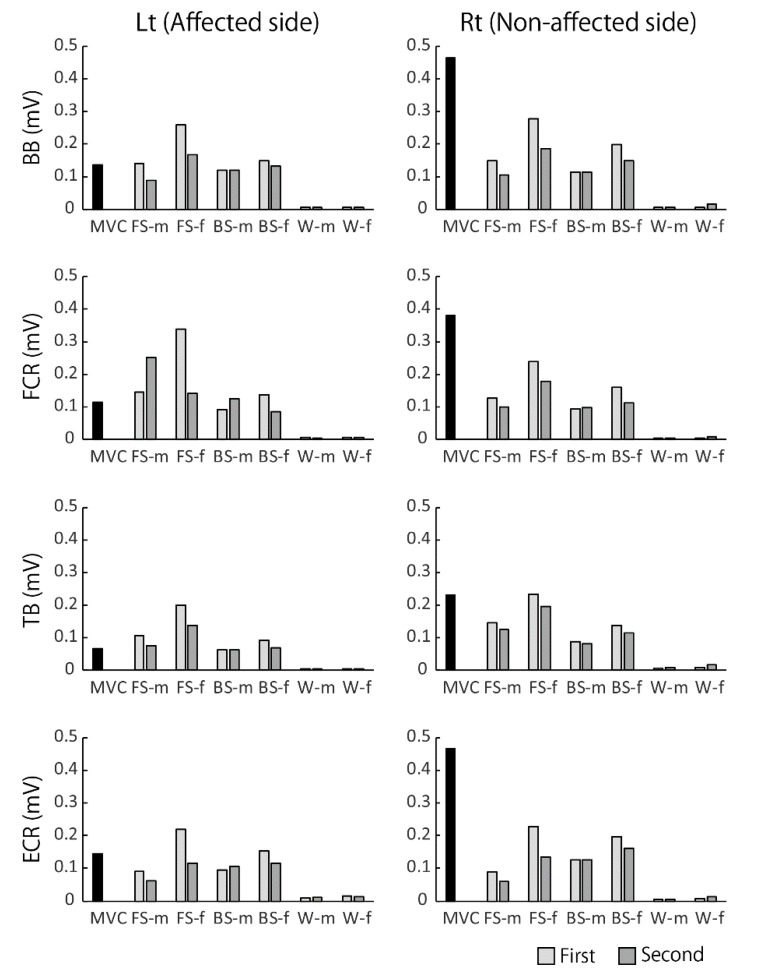
Comparisons of EMG activity levels during different motor tasks with the MVC EMG levels in the upper limb muscles. In the affected (left) side muscles, the EMG root square mean (RMS) values were comparable to or even greater in the faster speed crawl than the MVC RMSs. MVC; maximum voluntary contraction, FS-m; free style at moderate speed, FS-f; free style at faster speed, BS-m; breast stroke at moderate speed, BS-f; breast stroke at faster speed, W-m; walking at moderate speed, and W-f; walking at faster speed.

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
