# Peer review of "“Paralympic Brain”. Compensation and Reorganization of a Damaged Human Brain with Intensive Physical Training"

_sports, 2020, doi:10.3390/sports8040046_

Round 1

Reviewer 1 Report

Dear Authors,

First of all please meet my congratulations to you for choose interesting topic for the investigation. It is unique and very demanding process to research disable people in so different environment and adapt equipment especially to work in water. On the other hand, your results present very important theory about role of physical activity and exercise especially in the water in long term rehabilitation process.

I like your idea but according to reviewer obligations please find some points which should be improved in your article to push it on higher level, from my point of view, of course.

Title

Interesting and show main idea. However, my suggestion is to put Paralympic brain in quotation marks (“Paralympic brain” - ….) because it is neologism or metaphor used to attract readers but need to be right “wrap up”. By the way – you used it in the aim of study correctly.

Please correct also punctuation’s mistakes with dash instead of dot at the end and in the middle of title.

Abstract

Congratulations!!!!  You are the First!

However, according to requirements of "scientific style" of writing my suggestion is to change accent - firstly what was the main scientific idea of the study and next type of research to realize it, etc.

e.g. The main aim of the study was.... and according to our knowledge it was the first research presenting answers on presented aim/questions, or so.

Key words:

Should give yourself a chance to wider show what is the most important areas in your article and wider words used in the title. E.g. you used Paralympic in the title but did not sport or Paralympic sport and it could be used here,  

Others propositions: case study, training, female (sex is very important to present in key words, abstract and Methods), EMG, swimming, walking, etc.

Introduction

In my opinion it is very general and not enough introduce to the main topic, and/or point out importance of rehabilitation to “motivate” brain to be more active what is especially important in the case of disable people – from birth or as a result of different accidents.

You should present wider some elements of rehabilitation’s scientific base used in other references -  to similar research or other important for this point theory. Please do not base your introduction the only on 2 references! E.g. It would be interesting to give a signal that somebody did research in cooperation with Paralympic and that training for such people is very important, giving very good therapeutic results and how new tech try to support rehabilitation's proces/work in cooperation of course among representative of different specializations. On the other hand it would be very good to present that nobody up to the moment did similar to your study observations and it is one of a reason to prepare this study. Once more please shortly wider introduction in more  references to strength your point.

Aim

Reporting is very important but to increase importance of your work please think to change this sentence and present in opposite side to such way: firstly what was scientific aim of study and next eventually more detailed questions or other information - who was the subject and what type of research, protocol etc., but general you will describe it in Methods.

e.g. The main aim of presented study were evaluation of brain-muscle connectivity as well as her neuromuscular activities both during swimming in water and walking, to provide new insight into neurorehabilitation studies, namely “Paralympic brain”. Or so.

Material and Methods

It should be “manual” which present in details who was Subject of your study, what protocol you used, how you did it – prepare, recording, data analysis, etc. Everything have to be clear to give a chance to repeat similar research to other scientists or rehabilitant / physiotherapist who would like to use your protocol and especially use the results to help other people, thanks to your observations described in article.

According to it. Please: wider characteristics of Subject. We know Her age and she is titled Paralympic Athlete but we have no idea about the level of her fitness and sport's history earlier. You should shortly describe Her daily / weekly physical activities, training capacities - on the land and / or in the water - what is very important in context of your research and to have appropriate base to compare results with other people in the future study. How often she has rehabilitation and sport training, what is volume of its, whether she has the only training in the water or also in gym or reheb. center, or mixed, etc.  Is she use or live in a wheelchair (handy or electric?) or maybe she use walk on crutches? It must be clear. In case study history of the Subject life is a key to other elements of analysis.

On the other hand you describe well the technical protocol what is also very important to analyze results and compare with other research now or in the future.

You also omit any methods you used in your study to analyze data. I know that case study has a little different roles but descriptive stat is also methods which should be presented as well software you used during this process.

Results

In my opinion it is very well prepared part of study. I wonder whether it is possible to try deeper compare different parts of your observation but probably it would be difficult and depends on opportunity to use data output however in the situation you have one sample it could be difficult.

Description of presented results and graphs are correct.

Discussion

It is the most important part of article and must be compatible to title, introduction, aim and results as well base on very good prepared review of other article in the topic you work.

In the introduction to discussion we found main aim and way of research interest. We also can find new info about Subject “ZZ” -  “ZZ since early childhood likely plays a major role in the observed…” You returned to ZZ childhood! It is very important to prepare deeper analysis of results! OK!

But it confirm that you should present such “historical” information about ZZ in Material and use it in other parts of work – especially in Discussion. Please do it!

In other parts you analyzed and try to interpret results in ZZ context. However, there is still lack of stronger confirmation of your finding base on more numerous numbers of references.

I know you did such research as First! But somebody else did probably other investigations about similar problems or work with people with similar health problems like ZZ. It demand of course more work and reading more articles but effect will be much better and position of your study will be stronger.

Please add some reference to confirm your point and discuss it wider with other study with your area of interest.

I assess your Discussion and skills to describe ZZ results very high but lack of references confirmed your findings or examples then your finding debunk earlier views mobilized me to suggest your Team to try once more prepare wider and more splendid Discussion.   

References

The selection of articles presented in this part is correct. However, in my opinion, it should be wider to improve importance of idea and findings you present in article you sent to assess. Please add some new references to present situation (8 references).

 Please meet my suggestions and comments as friendly and positive. I’ve started read your text with high interest and I still feel it is great and not easy to realize work. Especially for it I would like to motivate you to harder work to have satisfaction and present this important results on highest possible level. I hope it will be impulse to change this version of text and in coming future I would finish reading your study with satisfaction and accept it to publish. Unfortunately, at present I can mark it very high and send to major revision. I am waiting for new version, soon.  

All the best to all your Team!

Reviewer 2 Report

The manuscript has written well in scientific and professional manner, however, there are few minor comments required to be corrected:

Page 1; Line 1: The results should be described in significant numeral values.

Page 2; Line 62: Should write the approval number.

Author Response

Thank you very much for your constructive comments. We changed our previous MS according to your comments and the other reviewers' comments.

Page 1; Line 1: The results should be described in significant numeral values.

We changed the previous sentences in Abstract and Results as follows.

Abstract:

Transcranial magnetic stimulation (TMS) to the affected and intact hand motor cortical area revealed that the affected side finger muscle cortical representation area shifted towards the temporal side, and cortico-spinal excitability of the target muscle was prominently facilitated, i.e., the maximum motor evoked potential in the affected side, 6.11±0.19 mV was greater than that in the intact side, 4.52±0.39 mV (mean±standard error).

Results:

Fig 2 shows the changes in MEP amplitude in relation to the applied TMS intensity (% of resting motor threshold, rMT) for both side FDI muscles. The results showed that the maximum slope of the sigmoidal curve (0.99 vs 0.28 mV/%rMT) and the maximum MEP amplitude (6.11±0.19 mV vs 4.52± 0.39 mV, mean±s.e.) were higher, and the motor threshold was lower (44 vs 48 % of maximum output of the device) in the affected side FDI compared with the non-affected side FDI.

Page 2; Line 62: Should write the approval number.

The approval number (475-4) was added to the sentence.

Reviewer 3 Report

Paper entitled “Paralympic brain -compensation and reorganization of a damaged human brain with intensive physical training” meets the necessary standards for publication in this journal.

I recommend: 

Please check the entire manuscript carefully for eventual typographical errors and language.
Final Conclusion: The paper meets the necessary standards for publication.

Author Response

Thank you very much for your comments. According to your comment we have thoroughly checked the revised MS, which we hope now can be accepted for the publication.

Reviewer 4 Report

This study aimed at evaluating brain-muscle connectivity and neuromuscular activities of a Paralympic swimmer. Although the study provided an interesting attempt to investigate a Paralympic athlete, there are several weaknesses which require further attention. As reported in the specific comments, several sections could be implemented.  

Specific comments

Introduction

Page 1, line 32-33. I suggest revising as follow “…to improve their performance and increase their competitive level.

Page 1, line 33-34. Please revise the entire sentence since it is not clear.

Page 1, line 35. Please revise “In other words, it can be said…”. It is not appropriate.

Page 1, line 37-38. Please revise accordingly “However, another major difference between athletic training and rehabilitation is the different level of motivation between athletes and patients.” Moreover, this sentence would require a reference to confirm it.

Page  1, line 40-43. If this sentence is not supported by previous study and there is not a reference, authors should state in a different way, making it more as hypothesis or speculation. Actually, the weakness of the introduction is the lack of emphasis on what is missing in literature and on this area of research. Moreover, a real research question is missing and after the purpose authors should formulate their own hypothesis.

Methods

Page 2, line 68. What does “both” refer to? Please clarify the sentence.

Methods and material section should be implemented since several information are missing.

Authors could clarify and provide more explanations about the rational for the evaluation of both walking and swimming activities.

How was the speed selected, standardized and controlled for both activities? Please clarify

Please provide more information about walking and swimming activities, like distance, duration.  

Did the authors collect information about the training history of the athlete?

A detailed data processing for electromyography has to be reported. Please provide more information.

Authors should better clarify how the data are reported, considering the complexity of the data. Also in results section.

Results

Figure 2. Could the authors clarify me the meaning of the white dots

Discussion

I suggest do not use CS abbreviation in discussion section.

Page 6, line 168. Considering that authors mentioned the importance of having training since early childhood, it will be important to retrieve and report information about training history of the athlete, like frequency, training load.

Page 7, line 177-187. Please spell out CST the first time.

Could authors provide more explanation about the possible mechanisms of motor cortical reorganization, please?

Page 7, line 213. Please revise “In an elegant series of studies…”. It is not appropriate. Please revise the entire sentence and make it more fluent.

At the end of discussion, authors could provide more practical implications from this study, even if is a case repo

Round 2

Reviewer 1 Report

Dear Authors,

I am very glad to find much better version of your manuscript and that my suggestions were accomplished.
In present form your work could bring important information which would be used in cooperation with other people with this same or similar healthy problems.

I hope it will be good motivation to work and wider this topic with benefit for others.

I wish you all Good and healthy on your professional and private way.

Reviewer 4 Report

The authors provided an extensive revision of the text, making it now suitable for publication. However, the quality of all figures must be improved, since it is not sufficient for publications.